# L-shaped association between the blood pressure response index and short-term mortality in intensive care patients with sepsis: An analysis based on the MIMIC-IV database

Heping Xu[1]*, Ruiyong Mo[2], Yiqiao Liu[2], Huan Niu[1], Xiongwei Cai[1], Ping He[1]

**1** Department of Emergency Medicine, Hainan General Hospital/Hainan Affiliated Hospital of Hainan Medical University, Haikou, Hainan, China, **2** Department of Emergency Medicine, Hainan Affiliated Hospital of Hainan Medical University, Haikou, Hainan, China

☯ These authors contributed equally to this work.
* xhp21528@163.com

## Abstract

### Background

Septic shock is a life-threatening condition that requires rapid assessment to reduce mortality. This study investigated the relationship between the blood pressure response index at admission and short-term mortality in septic shock patients.

### Methods

We retrospectively analyzed clinical data from the MIMIC-IV database via multivariable logistic regression, sensitivity analysis, and restricted cubic spline models to explore the associations between the BPRI at ICU admission and short-term mortality. Kaplan-Meier estimates and ROC curves evaluated BPRI's predictive value.

### Results

In 7,382 patients, a higher BPRI was linked to reduced short-term mortality. Each standard deviation increase in the BPRI led to 2.5%, 2.0%, and 1.8% reductions in in-hospital, 28-day, and 90-day mortality, respectively. K–M analysis revealed significantly lower short-term mortality in patients with higher BPRIs (P < 0.0001). An L-shaped nonlinear relationship between the BPRI and mortality was observed in the RCS model. The ROC curve for the BPRI showed an AUC of 0.640. Sensitivity analyses revealed an association even after patients with myocardial infarction, malignant cancer, or missing data were excluded.

**Data availability statement:** The datasets analyzed in this study are available in the MIMIC-IV database (https://mimic.physionet.org/), an anonymized resource approved by the IRBs of the Massachusetts Institute of Technology (MIT) and Beth Israel Deaconess Medical Center (BIDMC), with informed consent waived. Due to the sensitive nature of the data, direct sharing is not permitted. Researchers can access and request the data through the PhysioNet platform (https://physionet.org/content/mimiciv) after completing registration and CITI training. The authors did not have any special access privileges that would not be available to other researchers. All users can access the data under the same conditions.

**Funding:** This work was supported by the Hainan Provincial Natural Science Foundation of China. Project 823RC560,821RC1118. The funders had no role in study design, data collection and analysis, decision to publish, or preparation of the manuscript.

**Competing interests:** The authors have declared that no competing interests exist.

## Conclusion

An elevated BPRI at ICU admission is significantly associated with reduced short-term mortality in septic shock patients, indicating its potential as an early prognostic tool for risk assessment.

## Introduction

Septic shock is a critical and often fatal manifestation of sepsis, characterized by life-threatening organ dysfunction resulting from a dysregulated host response to infection [1]. It is one of the leading causes of mortality in intensive care unit (ICU) patients, with over a quarter of septic shock patients dying despite intensive treatment [2,3]. One of the key components of septic shock management is the use of vasoactive drugs, which aim to restore hemodynamic stability. However, the physiological mechanisms underlying the cardiovascular responses to vasoactive therapy in septic patients remain incompletely understood, limiting the ability to predict clinical outcomes and guide therapeutic decisions [1].

Sepsis is known to induce a complex cascade of changes in vascular tone, leading to hypotension and impaired tissue perfusion. Although cardiac output may remain stable in the early stages of septic shock, dysregulation of arterial tone leads to uneven blood flow distribution, resulting in suboptimal organ perfusion and contributing to multi-organ dysfunction [4,5]. Evaluating arterial tone is crucial for understanding these mechanisms, and several approaches have been proposed. For instance, the Windkessel model uses the ratio of left ventricular end-systolic pressure (LVESP) to stroke volume (SV) as an index of effective arterial elastance [6], while pulse contour analysis provides a method for assessing LVESP [7]. These techniques, however, are invasive and not well-suited for routine clinical use, particularly in settings where rapid, real-time assessments of vascular reactivity are needed.

Recent studies have sought to identify more accessible and non-invasive methods for assessing vascular function in septic shock. One promising approach is the Blood Pressure Reactivity Index (BPRI), introduced by Yujie Chen et al. [8]. The BPRI is calculated by dividing the mean arterial pressure (MAP) at a given time point by the Vasoactive-Inotropic Score (VIS) and is considered a potential predictor of prognosis in septic shock patients. However, the original definition of BPRI uses the vasoactive-inotropic score, which does not account for all vasoactive drugs commonly used in clinical practice, such as phenylephrine. This limitation restricts the BPRI's clinical utility and its ability to fully reflect the physiological responses to vasoactive therapy.

The primary mechanism of phenylephrine is the activation of α-adrenergic receptors, which promotes vasoconstriction, thereby increasing vascular resistance and blood pressure to improve the perfusion of vital organs. As a selective α1 receptor agonist, phenylephrine mainly acts on larger small arteries with minimal effects on terminal arterioles [9]. Additionally, phenylephrine helps in regulating heart rate [10,11] and can effectively reverse hemodynamic and metabolic abnormalities [12,13]. Although the 2021 Surviving Sepsis Campaign guidelines strongly

recommend norepinephrine as the first-choice vasopressor for septic shock [14], clinicians often avoid using norepinephrine in practice, particularly when septic patients develop rapid arrhythmias, due to its β-1 agonist effects, which may induce tachyarrhythmias [11]. Therefore, phenylephrine, as an alternative agent, may hold significant value in clinical applications for controlling heart rate and improving hemodynamics. Based on these considerations, we propose a refined definition of the Blood Pressure Response Index. The new definition incorporates the effects of vasopressor agents like phenylephrine, calculating the BPRI as the ratio of the mean arterial pressure to the vasopressor-inotrope score at a specific time point. This revised BPRI index can comprehensively reflect the combined effects of various vasopressor agents (including phenylephrine), providing a more accurate assessment of vascular reactivity.

The aim of this study is to investigate the relationship between the newly proposed BPRI at ICU admission and key clinical outcomes, including in-hospital mortality, 28-day mortality, and 90-day mortality, in patients with septic shock. We hypothesize that the revised BPRI will be a reliable predictor of short-term mortality, helping to guide clinical decision-making and improve patient prognostication in septic shock.

## Methods

### Data source

The data used in this study were sourced from the Medical Information Mart for Intensive Care IV (MIMIC-IV version 2.2) database [15,16], a registry developed by the Complex Systems Monitoring Group at Beth Israel Deaconess Medical Center (BIDMC) in Boston, Massachusetts. This dataset includes detailed records of over 50,000 patients admitted between 2008 and 2019. The Institutional Review Board at BIDMC approved the waiver of informed consent and authorized the sharing of research resources. Data extraction was conducted via PostgreSQL tools by the corresponding author, Heping Xu, who completed the CITI Program online training course (Record ID 59568270).

MIMIC-IV is an anonymized public database approved by the institutional review boards of the Massachusetts Institute of Technology (MIT) and Beth Israel Deaconess Medical Center (BIDMC). The requirement for informed consent was waived due to the thorough anonymization and de-identification of all patient information in the database.

### Definitions

The blood pressure response index was calculated by dividing the mean arterial pressure at a given time point by the vasoactive-inotropic score [8].

MAP [17]= (systolic blood pressure + 2 × diastolic blood pressure)/3. VIS was calculated as follows: dose of dopamine (µg/kg/min) + dobutamine (µg/kg/min) +epinephrine (µg/kg/min) × 100 + norepinephrine (µg/kg/min) ×100 + vasopressin (U/kg/min) ×10,000 + milrinone (µg/kg/min) × 10 + phenylephrine (µg/kg/min) ×10. The BPRI index was divided into four quartiles: Q1 (<5.60), Q2 (5.60--11.65), Q3 (11.65--21.55), and Q4 (>21.55). The diagnosis of sepsis was based on the Sepsis-3 criteria, defined as a Sequential Organ Failure Assessment (SOFA) score of 2 or more in the presence of infection. Septic shock was defined as the need for vasopressors in the context of sepsis with a lactate level greater than 2.0 mmol/L [1]. Acute kidney injury (AKI) is defined as an increase in serum creatinine of ≥ 0.3 mg/dL (≥ 26.5 µmol/L) within 48 hours, an increase in serum creatinine to ≥ 1.5 times the baseline level within the past week, or a reduction in urine output to < 0.5 mL/kg/h for 6 hours [18].

### Inclusion and exclusion criteria

Inclusion criteria:

1. Patients aged 18 years or older.

2. Patients were diagnosed with septic shock upon first ICU admission.

3. Patients with septic shock who have received at least one vasoactive drug (dopamine, dobutamine, epinephrine, phenylephrine, norepinephrine, vasopressin, or milrinone) within the first 24 hours of diagnosis.

Exclusion criteria:

1. Patients with prior ICU admissions were included to avoid data duplication.

2. Patients with a survival time of less than 48 hours were included to ensure sufficient evaluation of their clinical status and outcomes.

3. Patients with incomplete data on the mean arterial pressure (MAP) and vasoactive-inotropic score (VIS) are crucial for accurately calculating the BPRI.

### Outcome

The primary outcomes were in-hospital mortality, 28-day all-cause mortality, and 90-day all-cause mortality.

### Data extraction

The extracted dataset comprised a comprehensive range of demographic and clinical variables, including age, sex, ethnicity, weight, and history of myocardial infarction, congestive heart failure, chronic pulmonary disease, diabetes without control, severe liver disease, cerebrovascular disease, malignant cancer, and renal disease. Additionally, the study incorporated initial SOFA scores, the simplified acute physiology scores II (SAPS II), and the Charlson comorbidity index, along with vital signs such as systolic blood pressure, diastolic blood pressure, mean arterial pressure, heart rate, respiratory rate, temperature, and pulse oximetry readings. Laboratory parameters, including white blood cell count, hemoglobin, platelets, anion gap, bicarbonate, chloride concentration, glucose, sodium, potassium, creatinine, blood urea nitrogen, calcium, and prothrombin time, were also included. Clinical interventions such as invasive ventilation, renal replacement therapy (RRT), and acute kidney injury were documented. Furthermore, the study tracked ICU length of stay and total hospital length of stay. All baseline data were collected within the first 24 hours of ICU admission.

### Statistical analysis

Continuous variables are reported as the means (standard deviations) or medians (interquartile ranges), whereas categorical variables are expressed as percentages. Baseline characteristics were evaluated across different BPRI categories, with categorical data analyzed via the chi-square test, normally distributed continuous data analyzed via one-way analysis of variance, and nonnormally distributed data analyzed via the Kruskal–Wallis H test.

This study employed multivariable logistic regression analysis to explore the relationships between the BPRI and in-hospital mortality, 28-day all-cause mortality, and 90-day all-cause mortality. To assess multicollinearity, variance inflation factor (VIF) values were utilized, with VIF values exceeding 10 indicating significant multicollinearity. Four distinct models were developed: Model 1 adjusted for age, sex, ethnicity, and weight; Model 2 further adjusted for variables in Model 1 along with myocardial infarction, congestive heart failure, cerebrovascular disease, chronic pulmonary disease, diabetes without control, severe liver disease, malignant cancer, and renal disease; Model 3 additionally adjusted for the Charlson comorbidity index, SOFA score, SAPSII score, systolic blood pressure, diastolic blood pressure, heart rate, respiratory rate, temperature, and SpO2; and Model 4 included adjustments from Model 3, further accounting for white blood cell count, hemoglobin, platelet count, anion gap, bicarbonate, chloride, glucose, sodium, potassium, creatinine, blood urea nitrogen, calcium, and prothrombin time, as well as invasive ventilation, renal replacement therapy (RRT), and acute kidney injury (AKI).

Subgroup analyses were conducted on the basis of factors such as age (<65 years and ≥ 65 years), sex, ethnicity, myocardial infarction, congestive heart failure, cerebrovascular disease, chronic pulmonary disease, diabetes without control, severe liver disease, malignant cancer, renal disease, AKI, RRT, and invasive ventilation.

Sensitivity analyses were performed to further validate our findings. We conducted logistic regression analyses on patient subgroups, including those excluding patients with myocardial infarction and those excluding patients with malignant cancer and severe liver disease. Additionally, analyses were conducted using data with all original missing values excluded.

Restricted cubic splines were employed to determine threshold values and visualize the nonlinear relationship between the ICU admission BPRI score and short-term mortality. Kaplan–Meier analysis was used to compare survival among ICU septic shock patients stratified by the BPRI and to assess the impact of the BPRI on in-hospital mortality, 28-day mortality, and 90-day mortality. The differences in survival curves across different strata were visualized via a heatmap.

Receiver operating characteristic (ROC) curves were used to evaluate the predictive value of the BPRI, white blood cell (WBC) count, SOFA score, and their combined indices for short-term mortality in ICU septic shock patients. Data analysis was conducted via R version 4.2.1 and Stata version 18.0. All tests were two-sided, with a p value of less than 0.05 considered statistically significant.

## Results

### Baseline characteristics of the participants

A total of 7,382 patients from the MIMIC-IV database met the inclusion criteria, as illustrated in Fig 1. Table 1 provides a comprehensive overview of the baseline characteristics of patients with BPRIs. The mean age of these patients was 67.4 years, with a standard deviation of 15.3 years, and approximately 58.8% were male. The BPRI was categorized into four quartiles: Q1 (<5.60), Q2 (5.60--11.65), Q3 (11.65--21.55), and Q4 (>21.55). No significant differences were detected among the groups in terms of myocardial infarction, malignant cancer, Charlson comorbidity index, systolic blood pressure, diastolic blood pressure, mean arterial pressure, or blood sodium levels (P > 0.05). Patients in the highest quartile (Q4) presented significantly greater body weight and an increased prevalence of congestive heart failure, chronic pulmonary disease, diabetes without control, renal disease, and AKI, as well as a greater requirement for RRT. Conversely, these patients were younger; had a lower incidence of cerebrovascular disease and malignant cancer; and had lower respiratory rates, hemoglobin levels, platelet counts, incidence rates of invasive ventilation, and mortality rates, including in-hospital, 28-day, and 90-day mortality.

### Association of the BPRI score with short-term mortality

The associations between the BPRI score and short-term mortality are presented in Table 2. Patients were categorized into four groups on the basis of their BPRI levels. We developed four different logistic regression models to assess the independent impact of the BPRI on short-term mortality in ICU septic shock patients. Logistic regression analysis revealed that the BPRI score was negatively correlated with the risk of in-hospital mortality, 28-day mortality, and 90-day mortality. In Model 1, after adjusting for age, sex, ethnicity, and weight, the adjusted odds ratios (ORs) were 0.978 (95% CI 0.974–0.982), 0.980 (95% CI 0.976–0.985), and 0.984 (95% CI 0.981–0.988), respectively. In Model 2, which was based on Model 1, additional adjustments were made for myocardial infarction, congestive heart failure, cerebrovascular disease, chronic pulmonary disease, diabetes without control, severe liver disease, malignant cancer, and renal disease. The adjusted ORs were 0.978 (95% CI 0.974--0.982), 0.980 (95% CI 0.976--0.984), and 0.984 (95% CI 0.980--0.987), respectively. In Model 3, further adjustments were made for the Charlson comorbidity index, SOFA score, SAPSII score, systolic blood pressure (SBP), diastolic blood pressure (DBP), heart rate, respiratory rate, temperature, and SpO2. The adjusted ORs were 0.978 (95% CI 0.974--0.983), 0.980 (95% CI 0.976--0.984), and 0.984 (95% CI 0.980--0.988), respectively. In Model 4, which was based on Model 3, additional adjustments were made for white blood cell count, hemoglobin, platelet count, anion gap, bicarbonate, chloride, glucose, sodium, potassium, creatinine, blood urea nitrogen, calcium, prothrombin time, invasive ventilation, RRT, and AKI. The adjusted ORs were 0.975 (95% CI 0.971–0.980), 0.980 (95%

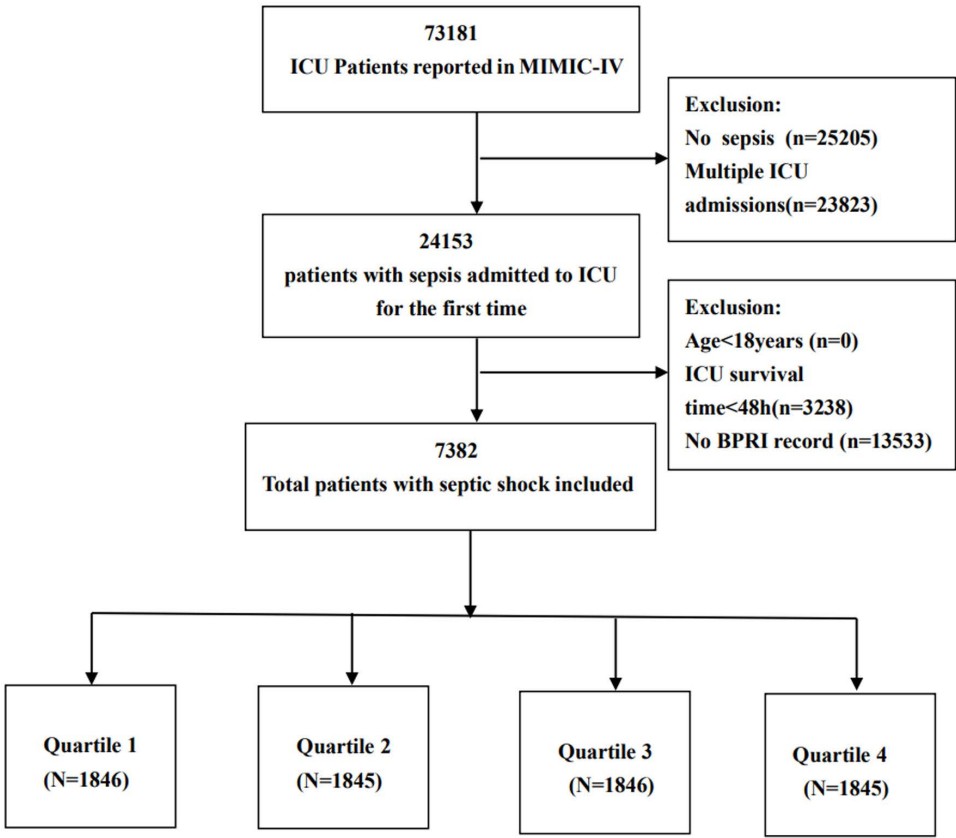

**Fig 1. Flow chart of patient selection for analysis.**

CI 0.975–0.984), and 0.982 (95% CI 0.978–0.986), respectively. The results of the logistic regression analysis, as shown in Table 2, indicate that, compared with the first quartile, the mortality risk significantly decreased in the second, third, and fourth quartiles.

## Restricted cubic spline

The threshold was determined via restricted cubic splines (RCSs) to visualize the nonlinear relationship between the ICU admission BPRI score and short-term mortality. As shown in Fig 2, the ICU admission BPRI score was nonlinearly and inversely correlated with in-hospital mortality, 28-day mortality, and 90-day mortality in patients with septic shock ($P < 0.0001$), forming an L-shaped curve. When the BPRI at admission is less than 11.6, the risk of in-hospital mortality increases rapidly as the BPRI decreases. Conversely, when the BPRI exceeds 11.6, the risk of in-hospital mortality gradually decreases as the BPRI increases (nonlinear $p < 0.001$). However, as the BPRI continues to increase, the rate of decline in the odds ratio (OR) slows. Overall, the OR for short-term mortality decreases as the BPRI level at ICU admission increases.

## Kaplan–Meier analysis

The study population was divided into four groups based on BPRI quartiles: Q1, Q2, Q3, and Q4. Kaplan–Meier analysis was then performed on the 28-day and 90-day mortality rates for patients with septic shock across these groups. As shown in Fig 3, the survival curve for the Q1 group was significantly lower than those for the Q2, Q3, and Q4 groups

**Table 1. Baseline characteristics of septic shock patients stratified by the BPRI quartiles.**

| Characteristics | | Q1 (N = 1846) | Q2 (N = 1845) | Q3 (N = 1846) | Q4 (N = 1845) | |
|---|---|---|---|---|---|---|
| BPRI | Overall(N = 7382) | BPRI <5.60 | 5.60 ≤ BPRI <11.65 | 11.65 ≤ BPRI<21.55 | 21.55 ≤ BPRI | P |
| Demographic | | | | | | |
| Age, years | 67.4 (15.3) | 67.3 (15.8) | 67.8 (15.0) | 67.1 (15.6) | 67.3 (14.7) | 0.008 |
| Sex (male, n) | 4340 (58.8) | 1055 (57.2) | 1099 (59.6) | 1135 (61.5) | 1051 (57.0) | 0.014 |
| Ethnicity (white, n) | 4902 (66.4) | 1175 (63.7) | 1251 (67.8) | 1260 (68.3) | 1216 (65.9) | 0.012 |
| Weight, kg | 84.0 (23.2) | 81.7 (23.0) | 82.5 (22.2) | 85.3 (23.4) | 86.6 (24.0) | 0.007 |
| Comorbidities | | | | | | |
| Myocardial infarction | 1403 (19.0) | 353 (19.1) | 353 (19.1) | 349 (18.9) | 348 (18.9) | 0.995 |
| Congestive heart failure | 2380 (32.2) | 613 (33.2) | 560 (30.4) | 563 (30.5) | 644 (34.9) | 0.006 |
| Chronic pulmonary disease | 1951 (26.4) | 494 (26.8) | 488 (26.4) | 449 (24.3) | 520 (28.2) | 0.065 |
| Diabetes without control | 1789 (24.2) | 411 (22.2) | 431 (23.4) | 460 (24.9) | 487 (26.4) | 0.020 |
| Severe liver disease | 491 (6.7) | 151 (8.2) | 114 (6.2) | 91 (4.9) | 135 (7.3) | <0.0001 |
| Cerebrovascular disease | 937 (13.4) | 237 (12.3) | 271 (14.7) | 225 (12.2) | 204 (11.1) | 0.009 |
| Malignant cancer | 873 (11.8) | 236 (12.8) | 235 (12.7) | 209 (11.3) | 193 (10.5) | 0.077 |
| Renal disease | 1696 (23.0) | 406 (22.0) | 365 (19.8) | 412 (22.3) | 513 (27.8) | <0.0001 |
| Severity scores | | | | | | |
| Charlson comorbidity index | 6 (4–8) | 6 (4–8) | 6 (4–8) | 6 (4–7) | 6 (4–8) | 0.056 |
| SOFA | 8 (5–11) | 9 (6–12) | 7 (5–10) | 7 (4–10) | 7 (5–11) | 0.0001 |
| SAPSII | 41 (33-51) | 45 (36-55) | 41 (32-51) | 39 (32–48) | 41 (33-52) | 0.0001 |
| Vital signs | | | | | | |
| SBP, mmHg | 111.2 (13.0) | 109.0 (12.7) | 110.8 (12.9) | 112.1 (12.7) | 113.1 (13.4) | 0.058 |
| DBP, mmHg | 59.4 (9.5) | 58.5 (9.5) | 59.1 (9.4) | 60.0 (9.6) | 60.0 (9.6) | 0.820 |
| MAP, mmHg | 76.7 (8.9) | 75.3 (8.8) | 76.3 (8.9) | 77.3 (8.8) | 77.7 (9.1) | 0.486 |
| Heart rate, beats/min | 87.1 (16.3) | 89.4 (16.9) | 86.7 (15.8) | 86.5 (15.9) | 85.8 (16.3) | 0.010 |
| Respiratory rate, beats/min | 19.9 (4.1) | 20.5 (4.3) | 19.7 (4.0) | 19.5 (4.0) | 19.7 (3.9) | <0.0001 |
| Temperature, °C | 36.9 (0.6) | 36.9 (0.7) | 36.9 (0.6) | 36.9 (0.6) | 36.8 (0.6) | <0.0001 |
| SpO2, % | 97.1 (2.2) | 97.0 (2.4) | 97.3 (2.0) | 97.3 (2.0) | 97.0 (2.4) | <0.0001 |
| Laboratory parameters | | | | | | |
| WBC, cell/mm3 | 12.7 (9.3-17.0) | 13.4 (9.7-18.6) | 12.8 (9.4-16.9) | 12.3 (9.0-16.1) | 12.4 (9.0-16.6) | 0.0001 |
| Hemoglobin, mg/dL | 10.3 (9.1-11.8) | 10.2 (9.0-11.7) | 10.3 (9.1-11.8) | 10.3 (9.1-11.8) | 10.2 (8.9-11.6) | 0.034 |
| Platelet, cell/mm3 | 179.0 (128.5-244.0) | 181.5 (127.5-257.5) | 180.5 (131.5-244.0) | 182 (131.5-241.5) | 173.0 (123.0-237.5) | 0.006 |
| Anion gap, mEq/L | 14.5 (12.0-17.0) | 15.5 (13.0-18.0) | 14.0 (12.0-17.0) | 14.0 (12.0-16.5) | 14.5 (12.0-17.0) | 0.0001 |
| Bicarbonate, mEq/L | 22.0 (19.5-24.5) | 21.5 (18.5-24.0) | 22.0 (19.5-24.5) | 22.5 (20.5-25.0) | 22.5 (20.0-25.0) | 0.0001 |
| Chloride, mEq/L | 105.0 (100.5-108.5) | 104.5 (100.0-108.5) | 105.5 (101.0-109.0) | 105.0 (101-108.5) | 105.0 (100.5-108.0) | 0.0008 |
| Glucose, mg/dL | 133.5 (111.5-169.0) | 139.0 (114.0-180.5) | 131.0 (109.5-162.0) | 132.0 (111.0-165.0) | 133.5 (111.0-168.0) | 0.0001 |
| Sodium, mEq/L | 138.5 (136-141) | 138.5 (135.5-141.0) | 138.5 (136.0-141.0) | 138.5 (136.0-141.0) | 138.5 (136.0-141.0) | 0.1568 |
| Potassium, mEq/L | 4.3 (3.9-4.7) | 4.3 (3.9-4.7) | 4.3 (3.9-4.7) | 4.2 (3.9-4.6) | 4.3 (3.9-4.7) | 0.0399 |
| Creatinine, mg/dL | 1.1 (0.8-1.8) | 1.3 (0.85-2.0) | 1.1 (0.8-1.6) | 1.1 (0.75-1.6) | 1.2 (0.8-2.0) | 0.0001 |
| BUN, mg/dL | 22.0 (15-37.5) | 25.0 (16.0-41.0) | 21.5 (14.5-37.0) | 20.5 (14.5-32.5) | 23.0 (15.5-38.5) | 0.0001 |
| Calcium, m Eq/L | 8.2 (7.8-8.6) | 8.1 (7.6-8.6) | 8.2 (7.8-8.6) | 8.2 (7.9-8.6) | 8.2 (7.8-8.6) | 0.0001 |
| PT, sec | 14.7 (13.1-17.4) | 15.0 (13.3-18.9) | 14.7 (13.0-17.1) | 14.5 (13.0-16.6) | 14.7 (13.2-17.3) | 0.0001 |
| Outcome | | | | | | |
| Invasive ventilation | 3108 (42.1) | 939 (50.9) | 777 (42.1) | 704 (38.1) | 688 (37.3) | <0.0001 |
| RRT | 1166 (15.8) | 312 (16.9) | 161 (8.7) | 216 (11.7) | 477 (25.9) | <0.0001 |
| AKI | 6529 (88.4) | 1645 (89.1) | 1590 (86.2) | 1616 (87.5) | 1678 (90.9) | <0.0001 |
| LOS ICU | 5.2 (3.2-9.9) | 5.8 (3.5-11.0) | 5.1 (3.2-9.2) | 4.9 (3.0-9.3) | 5.2 (3.1-10.6) | 0.0001 |

*(Continued)*

**Table 1.** (Continued)

| Characteristics | | Q1 (N = 1846) | Q2 (N = 1845) | Q3 (N = 1846) | Q4 (N = 1845) | |
|---|---|---|---|---|---|---|
| LOS hospital | 10.9 (6.8-18.1) | 10.7 (6.3-17.7) | 10.9 (6.9-17.8) | 10.4 (6.9-17.8) | 11.6 (7.1-19.8) | 0.0001 |
| In-hospital mortality | 1772 (24.0) | 741 (40.1) | 401 (21.7) | 319 (17.3) | 311 (16.8) | <0.0001 |
| 28-day mortality | 1917 (26.0) | 780 (42.3) | 433 (23.5) | 358 (19.4) | 346 (18.8) | <0.0001 |
| 90-day mortality | 2431 (32.9) | 898 (48.6) | 579 (31.4) | 476 (25.8) | 478 (25.9) | <0.0001 |

Continuous variables are presented as the means (SDs) or medians (quartiles), whereas categorical variables are presented as absolute numbers (percentages). SBP, systolic blood pressure; DBP, diastolic blood pressure; MAP, mean artery pressure; SpO2, pulse oxygen saturation; SOFA, Sequential Organ Failure Assessment score; SAPS II, simplified acute physiology score II; WBC, white blood cell; BUN, blood urea nitrogen; RRT, renal replacement therapy; AKI, acute kidney injury; BPRI, blood pressure response index; LOS, length of stay.

among ICU-admitted patients with septic shock (log-rank test, P < 0.0001). Significant differences were observed between all groups except between Q3 and Q4 (P < 0.05), as shown in Fig 4. Thus, a lower BPRI upon ICU admission is associated with increased 28-day and 90-day mortality.

### Receiver operating characteristic curve analysis

We conducted receiver operating characteristic (ROC) curve analysis to evaluate the predictive value of the BPRI for short-term mortality in ICU septic shock patients and compared its performance with that of the WBC count, SOFA score, and their combined predictions (Fig 5). The areas under the curve (AUCs) for the BPRI model in predicting in-hospital mortality, 28-day mortality, and 90-day mortality were 0.64 (95% CI: 0.62–0.65), 0.63 (95% CI: 0.61–0.65), and 0.61 (95% CI: 0.60–0.63), respectively, which were significantly greater than those of the WBC model [0.53 (95% CI: 0.51–0.55), 0.54 (95% CI: 0.53–0.56), and 0.52 (95% CI: 0.51–0.54); all P < 0.001] but lower than those of the SOFA model. Additionally, the combined BPRI and SOFA model outperformed the combined BPRI and WBC model, as did the other models (Table 3). Therefore, the BPRI combined with the SOFA score upon admission has good predictive value for short-term mortality in patients with septic shock.

### Subgroup analysis

To explore potential clinical heterogeneity, we conducted interaction and stratification analyses (Fig 6). We assessed the associations between the BPRI score and in-hospital mortality, 28-day mortality, and 90-day mortality across different subgroups. The stratification analyses were based on age (<65 years and ≥65 years), sex, ethnicity, myocardial infarction, congestive heart failure, cerebrovascular disease, chronic pulmonary disease, diabetes without control, severe liver disease, malignant cancer, renal disease, AKI, RRT, and invasive ventilation. Significant interactions were observed only for cerebrovascular disease, severe liver disease, and RRT (P < 0.05).

### Sensitivity analysis

The results of the sensitivity analysis are presented in Table 4. After patients with myocardial infarction were excluded, the odds ratios (ORs) for in-hospital mortality, 28-day mortality, and 90-day mortality were 0.977 (95% CI: 0.972–0.982), 0.979 (95% CI: 0.975–0.984), and 0.981 (95% CI: 0.977–0.986), respectively. When individuals with both malignant cancer and myocardial infarction were excluded, the ORs were 0.977 (95% CI: 0.972–0.983), 0.980 (95% CI: 0.974–0.985), and 0.981 (95% CI: 0.976–0.986), respectively. After all individuals with missing values were excluded, the ORs were 0.976 (95% CI: 0.971--0.981), 0.981 (95% CI: 0.977--0.986), and 0.983 (95% CI: 0.979--0.988), respectively. The trend test was also statistically significant across the Q1, Q2, Q3, and Q4 stratifications (P < 0.0001).

**Table 2. Relationships between the BPRI and clinical outcomes in different models.**

| BPRI | Model 1 | | Model 2 | | Model 3 | | Model 4 | |
|---|---|---|---|---|---|---|---|---|
| | OR (95% CI) | P value | OR (95% CI) | P value | OR (95% CI) | P value | OR (95% CI) | P value |
| In-hospital mortality | 0.978 (0.974, 0.982) | <0.0001 | 0.978 (0.974, 0.982) | <0.0001 | 0.978 (0.974, 0.983) | <0.0001 | 0.975 (0.971, 0.980) | <0.0001 |
| Quartile | | | | | | | | |
| Quartile 1 | Ref | | Ref | | Ref | | Ref | |
| Quartile 2 | 0.413 (0.357, 0.478) | <0.0001 | 0.409 (0.352, 0.474) | <0.0001 | 0.474 (0.405, 0.555) | <0.0001 | 0.518 (0.437, 0.613) | <0.0001 |
| Quartile 3 | 0.315 (0.270, 0.367) | <0.0001 | 0.321 (0.274, 0.375) | <0.0001 | 0.395 (0.334, 0.466) | <0.0001 | 0.412 (0.345, 0.492) | <0.0001 |
| Quartile 4 | 0.300 (0.257, 0.351) | <0.0001 | 0.293 (0.250, 0.343) | <0.0001 | 0.305 (0.257, 0.361) | <0.0001 | 0.280 (0.233, 0.337) | <0.0001 |
| P for trend | <0.0001 | | <0.0001 | | <0.0001 | | <0.0001 | |
| 28-day mortality | 0.980 (0.976, 0.985) | <0.0001 | 0.980 (0.976, 0.984) | <0.0001 | 0.980 (0.976, 0.984) | <0.0001 | 0.980 (0.975, 0.984) | <0.0001 |
| Quartile 1 | Ref | | Ref | | Ref | | Ref | |
| Quartile 2 | 0.415 (0.359, 0.479) | <0.0001 | 0.408 (0.352, 0.473) | <0.0001 | 0.467 (0.400, 0.546) | <0.0001 | 0.520 (0.441, 0.613) | <0.0001 |
| Quartile 3 | 0.331 (0.285, 0.385) | <0.0001 | 0.337 (0.289, 0.392) | <0.0001 | 0.406 (0.346, 0.478) | <0.0001 | 0.445 (0.375, 0.528) | <0.0001 |
| Quartile 4 | 0.315 (0.270, 0.366) | <0.0001 | 0.305 (0.262, 0.356) | <0.0001 | 0.317 (0.269, 0.374) | <0.0001 | 0.321 (0.269, 0.383) | <0.0001 |
| P for trend | <0.0001 | | <0.0001 | | <0.0001 | | <0.0001 | |
| 90-day mortality | 0.984 (0.981, 0.988) | <0.0001 | 0.984 (0.980, 0.987) | <0.0001 | 0.984 (0.980, 0.988) | <0.0001 | 0.982 (0.978, 0.986) | <0.0001 |
| Quartile 1 | Ref | | Ref | | Ref | | Ref | |
| Quartile 2 | 0.475 (0.415, 0.545) | <0.0001 | 0.466 (0.405, 0.537) | <0.0001 | 0.543 (0.467, 0.631) | <0.0001 | 0.607 (0.518, 0.711) | <0.0001 |
| Quartile 3 | 0.368 (0.319, 0.423) | <0.0001 | 0.370 (0.320, 0.428) | <0.0001 | 0.452 (0.387, 0.527) | <0.0001 | 0.490 (0.416, 0.577) | <0.0001 |
| Quartile 4 | 0.369 (0.320, 0.424) | <0.0001 | 0.351 (0.304, 0.407) | <0.0001 | 0.373 (0.319, 0.436) | <0.0001 | 0.361 (0.305, 0.427) | <0.0001 |
| P for trend | <0.0001 | | <0.0001 | | <0.0001 | | <0.0001 | |

OR, odds ratio; CI, confidence interval; Ref, reference; Model 1: Adjusted for age, sex, ethnicity and weight. Model 2: Adjusted for variables included in Model 1+myocardial infarction, congestive heart failure, cerebrovascular disease, chronic pulmonary disease, diabetes without control, severe liver disease, malignant cancer and renal disease.

Model 3: Adjusted for variables included in Model 2+Charlson comorbidity index, SOFA score, SAPSII score, SBP, DBP, MAP, heart rate, respiratory rate, temperature and SpO2. Model 4: Adjusted for variables included in Model 3+white blood cell count, hemoglobin, platelet count, anion gap, bicarbonate, chloride, glucose, sodium, potassium, creatinine, blood urea nitrogen, calcium and prothrombin time, invasive ventilation, RRT and AKI.

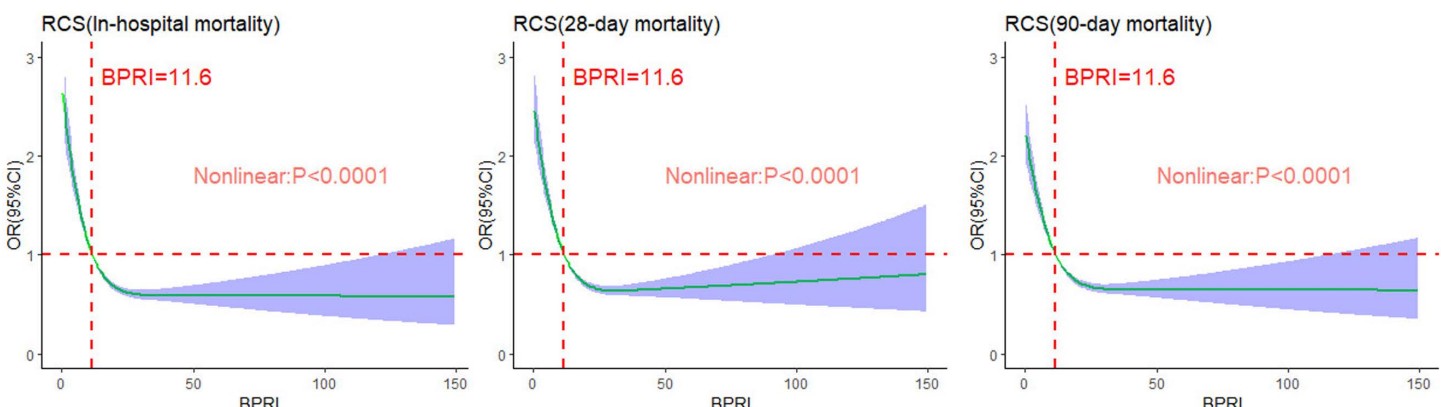

**Fig 2. Nonlinear relationship between the BPRI and short-term mortality.**

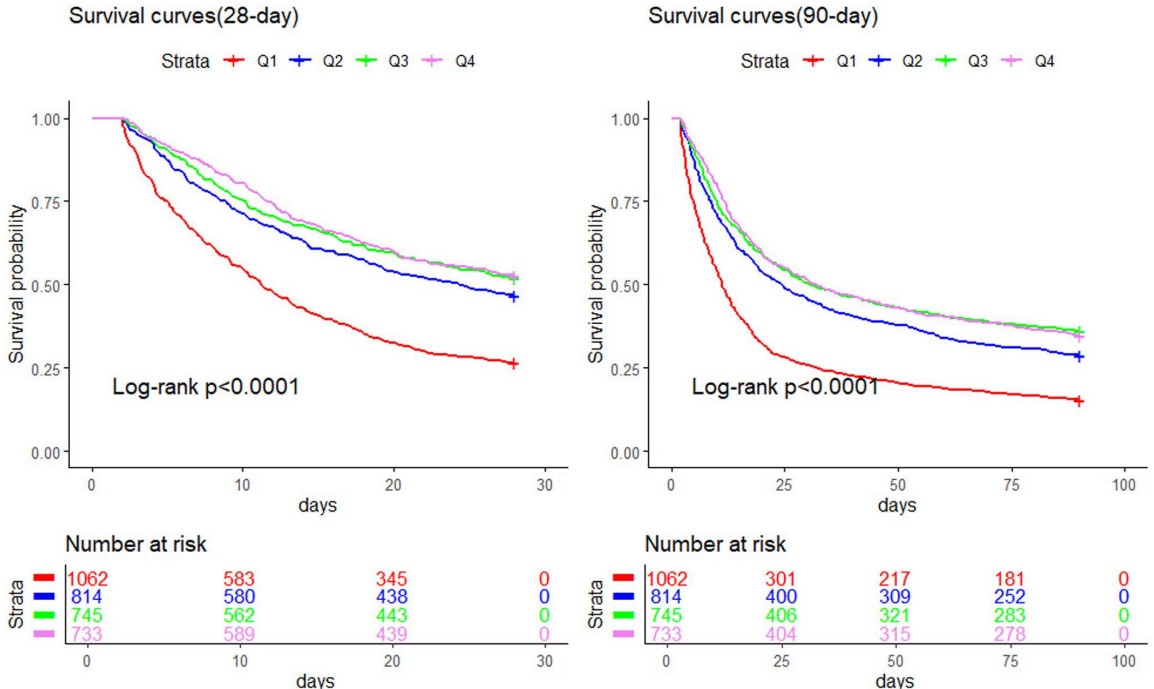

**Fig 3. Kaplan–Meier plots for short-term mortality by ICU admission BPRI strata.**

## Discussion

This study introduces a redefined BPRI, which is calculated as the ratio of the MAP to the VIS. The BPRI is proposed as a measure to assess the responsiveness of septic shock patients to vasoactive drugs. The primary objective was to explore the potential role of the BPRI in predicting short-term mortality among these patients.

The findings revealed a significant inverse, nonlinear association between the BPRI at ICU admission and short-term mortality, characterized by an L-shaped curve. Specifically, patients with lower BPRIs at admission had substantial increases in in-hospital, 28-day, and 90-day mortality, whereas those with higher BPRIs had better survival outcomes.

K–M survival analysis confirmed these results, showing that the survival curve for the lowest BPRI quartile group was significantly lower than that for the other three groups, particularly during the 28-day and 90-day follow-up periods. Moreover, ROC curve analysis indicated that combining the BPRI score with the SOFA score improved the accuracy of the prediction of short-term mortality in septic shock patients. This suggests that while the BPRI alone may have limited predictive power, its prognostic value is significantly enhanced when it is integrated with other clinical scoring systems.

Subgroup analysis further revealed that the BPRI had a more pronounced protective effect in specific patient populations, such as those with severe liver disease or those receiving renal replacement therapy. These findings suggest that patients with these conditions are more vulnerable to hemodynamic instability, making the BPRI a particularly valuable prognostic tool in these patients. Sensitivity analysis confirmed that even after patients with specific conditions were excluded, the association between the BPRI score and short-term mortality remained robust, further validating the potential of the BPRI score as a tool for clinical assessment.

Previous studies, including those by Gordon et al., have established a link between reduced responsiveness to vasoactive drugs and increased mortality in patients with septic shock [19]. However, while the importance of vascular reactivity has been recognized, a simple, feasible, and real-time assessment method is lacking. Earlier studies, such as those by

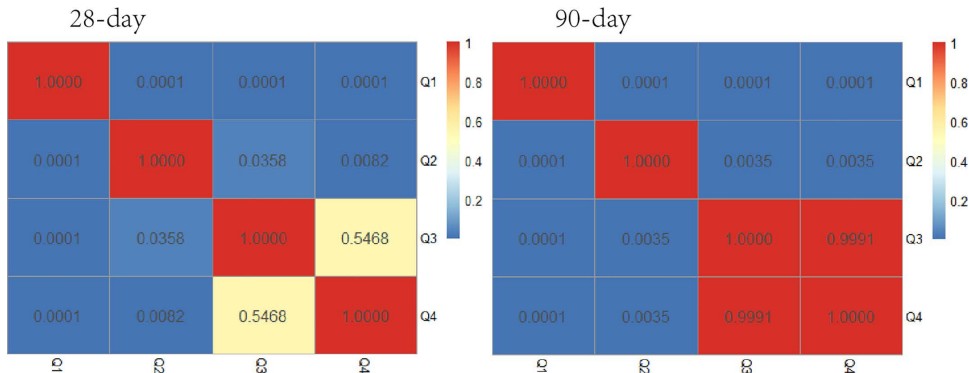

**Fig 4. Heatmap of differences in survival curves.**

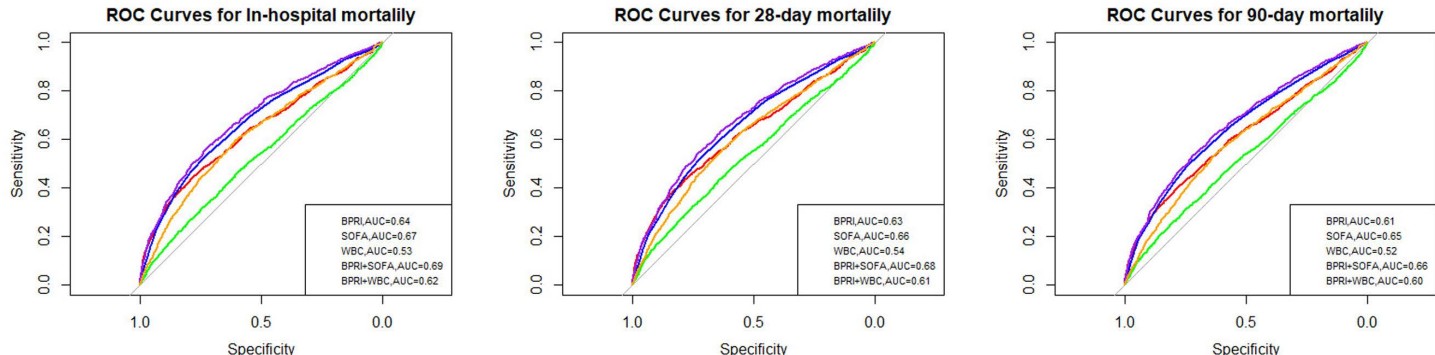

**Fig 5. Receiver operating characteristic curves for evaluating the predictive value of the BPRI, WBC count and SOFA score for the short-term mortality of septic shock patients in the ICU.**

**Table 3. Area under the curve across different models.**

| Model | BPRI | SOFA | WBC | BPRI+SOFA | BPRI+WBC |
|---|---|---|---|---|---|
| | AUC 95%CI | AUC 95%CI | AUC 95%CI | AUC 95%CI | AUC 95%CI |
| In-hospital mortality | 0.64 (0.62, 0.65) | 0.67 (0.66, 0.68) | 0.53 (0.51, 0.55) | 0.69 (0.67, 0.70) | 0.62 (0.60, 0.63) |
| 28-day mortality | 0.63 (0.61, 0.65) | 0.66 (0.65, 0.67) | 0.54 (0.53, 0.56) | 0.68 (0.66, 0.69) | 0.61 (0.60, 0.63) |
| 90-day mortality | 0.61 (0.60, 0.63) | 0.65 (0.64, 0.67) | 0.52 (0.51, 0.54) | 0.66 (0.65, 0.68) | 0.60 (0.58, 0.61) |

Monge Garcia MI et al., explored dynamic arterial elasticity (Eadyn), the ratio of pulse pressure variation (PPV) to stroke volume variation (SVV). However, these studies were limited mainly to animal models and did not fully capture the clinical complexity needed for real-world application [20,21]. Eadyn's measurements are influenced by various factors, including the respiratory cycle and ventriculo-arterial coupling [22,23], limiting its ability to reflect arterial elasticity and the overall impact of vasoactive drugs [23–25]. While adrenergic drugs are commonly used for hemodynamic stabilization in septic shock, their effects vary significantly depending on the patient's condition [26–28].

To address these limitations, our study aimed to develop a dynamic and clinically relevant assessment of vasoactive drug responsiveness. Building on the work of Yujie Chen et al. [8], we improved the VIS scoring

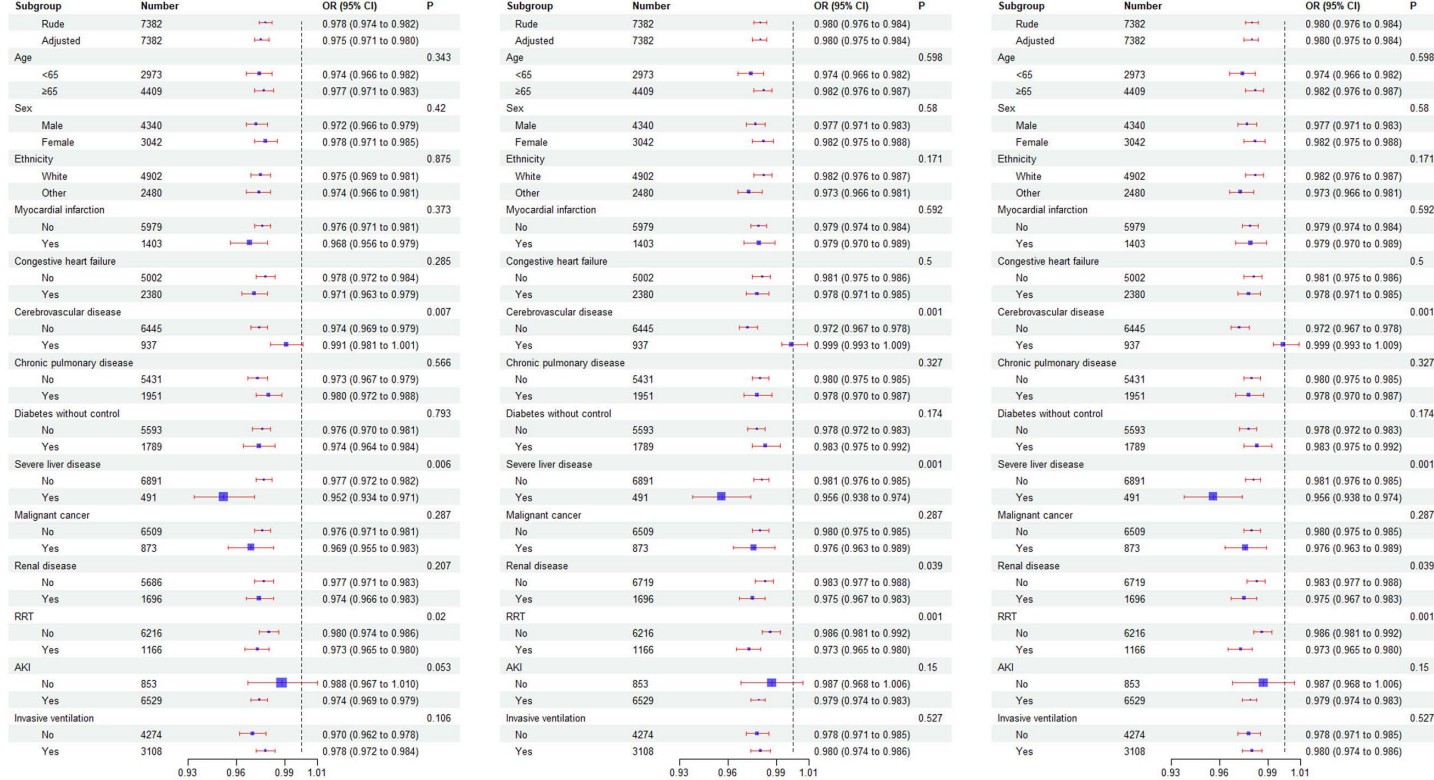

**Fig 6. Effect size of the BPRI on in-hospital mortality, 28-day mortality, and 90-day mortality across prespecified and exploratory subgroups:** The effect sizes were analyzed across subgroups on the basis of variables such as age, sex, ethnicity and weight, myocardial infarct, congestive heart failure, cerebrovascular disease, chronic pulmonary disease, diabetes without control, severe liver disease, malignant cancer and renal disease, Charlson comorbidity index, SOFA score, SAPSII score, SBP, DBP, MAP, heart rate, respiratory rate, temperature and SpO2, white blood cell count, hemoglobin, platelet count, anion gap, bicarbonate, chloride, glucose, sodium, potassium, creatinine, blood urea nitrogen, calcium and prothrombin time, invasive ventilation, RRT and AKI, with the exception of the subgroup variable.

algorithm and observed a significant negative correlation between the BPRI score and in-hospital mortality. This finding suggests that establishing a critical BPRI threshold could aid in risk stratification and enable rapid bedside assessment of patient prognosis. Specifically, a BPRI less than 11.6 was associated with a significantly increased risk of in-hospital mortality, indicating the need for more aggressive treatment in these patients. The ability of the BPRI to reflect the impact of vasoactive drugs on blood pressure promptly provides an early warning of mortality risk, and its performance in predicting early mortality has remained stable. When combined with SOFA scores, the BPRI provides unique, actionable information that is suitable for clinical implementation.

The relationship between the BPRI and mortality revealed in this study may reflect the vascular system's response to infection in septic shock patients. Septic shock is often associated with severe vasodilation and hypotension, which are believed to result from the abnormal release of inflammatory mediators such as nitric oxide, prostaglandins, and cytokines [29]. These mediators cause vascular smooth muscle relaxation and increased vascular permeability, leading to uneven blood distribution and insufficient organ perfusion [30], as well as reduced blood volume and refractory edema [31,32]. As an index that combines the MAP and VIS, the BPRI may effectively capture these pathophysiological changes, making it a powerful prognostic indicator. Through a large sample analysis, this study further validated the predictive value of the BPRI.

**Table 4. Sensitivity analyses.**

| BPRI | In-hospital mortality | | | 28-day mortality | | | 90-day mortality | | |
|---|---|---|---|---|---|---|---|---|---|
| | OR (95%CI) | P | P for trend | OR (95%CI) | P | P for trend | OR (95%CI) | P | P for trend |
| Excluding participants with myocardial infarction | | | | | | | | | |
| Model | 0.977 (0.972, 0.982) | <0.0001 | | 0.979 (0.975, 0.984) | <0.0001 | | 0.981 (0.977, 0.986) | <0.0001 | |
| Quartile 1 | Ref | | | Ref | | | Ref | | |
| Quartile 2 | 0.504 (0.417, 0.608) | <0.0001 | | 0.527 (0.439, 0.633) | <0.0001 | | 0.600 (0.503, 0.716) | <0.0001 | |
| Quartile 3 | 0.416 (0.341, 0.506) | <0.0001 | | 0.454 (0.375, 0.549) | <0.0001 | | 0.488 (0.407, 0.585) | <0.0001 | |
| Quartile 4 | 0.283 (0.230, 0.348) | <0.0001 | <0.0001 | 0.320 (0.262, 0.390) | <0.0001 | <0.0001 | 0.365 (0.303, 0.440) | <0.0001 | <0.0001 |
| Excluding participants with myocardial infarction and malignant cancer | | | | | | | | | |
| Model | 0.977 (0.972, 0.983) | <0.0001 | | 0.980 (0.974, 0.985) | <0.0001 | | 0.981 (0.976, 0.986) | <0.0001 | |
| Quartile 1 | Ref | | | Ref | | | Ref | | |
| Quartile 2 | 0.479 (0.389, 0.591) | <0.0001 | | 0.509 (0.415, 0.623) | <0.0001 | | 0.548 (0.451, 0.666) | <0.0001 | |
| Quartile 3 | 0.389 (0.313, 0.483) | <0.0001 | | 0.405 (0.328, 0.500) | <0.0001 | | 0.436 (0.357, 0.533) | <0.0001 | |
| Quartile 4 | 0.278 (0.222, 0.348) | <0.0001 | <0.0001 | 0.306 (0.246, 0.380) | <0.0001 | <0.0001 | 0.330 (0.268, 0.405) | <0.0001 | <0.0001 |
| Exclude all individuals with missing values | | | | | | | | | |
| Model | 0.976 (0.971, 0.981) | <0.0001 | | 0.981 (0.977, 0.986) | <0.0001 | | 0.983 (0.979, 0.988) | <0.0001 | |
| Quartile 1 | Ref | | | Ref | | | Ref | | |
| Quartile 2 | 0.540 (0.451, 0.647) | <0.0001 | | 0.523 (0.439, 0.624) | <0.0001 | | 0.622 (0.524, 0.737) | <0.0001 | |
| Quartile 3 | 0.414 (0.342, 0.502) | <0.0001 | | 0.443 (0.369, 0.532) | <0.0001 | | 0.496 (0.416, 0.592) | <0.0001 | |
| Quartile 4 | 0.295 (0.242, 0.359) | <0.0001 | <0.0001 | 0.337 (0.279, 0.407) | <0.0001 | <0.0001 | 0.383 (0.320, 0.459) | <0.0001 | <0.0001 |

OR, odds ratio; CI, confidence interval; Ref, reference; Adjusted for age, sex, ethnicity and weight, myocardial infarct, congestive heart failure, cerebro-vascular disease, chronic pulmonary disease, diabetes without control, severe liver disease, malignant cancer and renal disease, Charlson comorbidity index, SOFA score, SAPSII score, SBP, DBP, heart rate, respiratory rate, temperature and SpO2, white blood cell count, hemoglobin, platelet count, anion gap, bicarbonate, chloride, glucose, sodium, potassium, creatinine, blood urea nitrogen, calcium and prothrombin time, invasive ventilation, RRT and AKI.

Despite the valuable insights provided by this study, several limitations should be noted. First, the research relies on retrospective database analysis, and while the sample size is large, it cannot fully control for confounding variables, potentially introducing selection bias. Second, factors such as fluid resuscitation, clinical interventions, and the complex clinical course of sepsis may introduce confounders that are difficult to control in observational studies. These factors highlight areas for future research. Additionally, this study did not account for other factors that might influence blood pressure responsiveness, such as disease progression and individual metabolic differences, which could affect the accuracy of the BPRI. Finally, since this study is primarily based on single-center data, further validation through multicenter studies is necessary to establish the generalizability of the results.

## Conclusion

This cohort study demonstrated that a higher BPRI is closely associated with a reduced short-term mortality rate in ICU patients with septic shock. These findings suggest that the BPRI could serve as a potential prognostic tool to aid clinicians in early risk assessment and timely intervention in patients with septic shock.

## Acknowledgments

We thank the MIMIC-IV database for providing the original study data.

## Author contributions

**Conceptualization:** Heping Xu, Yiqiao Liu.

**Data curation:** Heping Xu, Ruiyong Mo, Yiqiao Liu.

**Formal analysis:** Heping Xu, Ruiyong Mo, Yiqiao Liu.

**Funding acquisition:** Heping Xu.

**Methodology:** Ping He.

**Visualization:** Heping Xu, Ruiyong Mo, Yiqiao Liu, Huan Niu, XiongWei Cai, Ping He.

**Writing – original draft:** Heping Xu, Ruiyong Mo, Yiqiao Liu.

**Writing – review & editing:** Huan Niu, XiongWei Cai, Ping He.

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
