## [Editor Report · Decision Letter 0]

8 Nov 2024

Dear Dr. Xu,

Thank you for submitting your manuscript to PLOS ONE. After careful consideration, we feel that it has merit but does not fully meet PLOS ONE’s publication criteria as it currently stands. Therefore, we invite you to submit a revised version of the manuscript that addresses the points raised during the review process.

We look forward to receiving your revised manuscript.

Kind regards,

Fadi Aljamaan

Academic Editor

PLOS ONE

Journal Requirements:

Additional Editor Comments :

Before proceeding with your paper for decision about revision, you need to enrich the introduction part and explain the rationale and the literature and evidence support physiological and by literature for the newly proposed definition used for the BPRI (MAP divided by  VIS)

---

## [Author Response · Author response to Decision Letter 1]

12 Nov 2024

Dear Dr. Aljamaan,

Thank you very much for your email and the constructive feedback on our manuscript entitled "L-Shaped Association between the Blood Pressure Response Index and Short-Term Mortality in Intensive Care Patients with Sepsis: An Analysis Based on the MIMIC-IV Database." We greatly appreciate the time and effort that you and the reviewers have dedicated to evaluating our work.

We are grateful for the opportunity to revise our manuscript and address the points raised during the review process. We have carefully reviewed and incorporated the suggestions provided by the reviewers and the editor to improve the manuscript, ensuring it meets the publication criteria of PLOS ONE. Specifically, we have included detailed responses to each point in the "Response to Reviewers" document, and the changes have been highlighted in red in the revised manuscript.

Once again, thank you for the attention and assistance from you and the reviewers. We look forward to your further feedback and hope that the revisions will enhance the quality of the manuscript.

Kind regards,

Heping Xu, PhD

Corresponding Author

Department of Emergency Medicine

Hainan General Hospital / Hainan Affiliated Hospital of Hainan Medical University

Haikou City, Hainan Province, 570311

Email: xhp21528@163.com

---

## [Editor Report · Decision Letter 1]

21 Nov 2024

Dear Dr. Xu,

Thank you for submitting your manuscript to PLOS ONE. After careful consideration, we feel that it has merit but does not fully meet PLOS ONE’s publication criteria as it currently stands. Therefore, we invite you to submit a revised version of the manuscript that addresses the points raised during the review process.

We look forward to receiving your revised manuscript.

Kind regards,

Fadi Aljamaan

Academic Editor

PLOS ONE

Additional Editor Comments:

Dear authors

You did not respond still to my query regarding the rationale and literature evidence of using this new definition of BPRI, you cited only one study from your group but this is new definition and you need to explain it physiologically, having significant statistical association does not prove the new concept

---

## [Author Response · Author response to Decision Letter 2]

10 Dec 2024

Dear Dr. Ramos,

Thank you for your thoughtful feedback on our "L-Shaped Association between the Blood Pressure Response Index and Short-Term Mortality in Intensive Care Patients with Sepsis: An Analysis Based on the MIMIC-IV Database." (Manuscript PONE-D-24-45268R1). We appreciate the time and effort the reviewers have taken to evaluate our work, and we are grateful for the opportunity to revise and resubmit the manuscript.

We are grateful for the opportunity to revise our manuscript and address the points raised during the review process. We will carefully review and incorporate the suggestions to improve the manuscript, ensuring that we meet PLOS ONE publication criteria.

We once again appreciate the opportunity to improve our manuscript. We look forward to submitting the revised version and hope it meets your expectations. Should further revisions be needed, please do not hesitate to let us know.

Kind regards,

Heping Xu, PhD

Department of Emergency Medicine

Hainan General Hospital / Hainan Affiliated Hospital of Hainan Medical University

Haikou City, Hainan Province, 570311

Email: xhp21528@163.com

December 10, 2024

---

## [Decision Letter · Decision Letter 2]

7 May 2025

L-Shaped Association between the Blood Pressure Response Index and Short-Term Mortality in Intensive Care Patients with Sepsis: An Analysis Based on the MIMIC-IV Database

PONE-D-24-45268R2

Dear Dr. Heping Xu,

We’re pleased to inform you that your manuscript has been judged scientifically suitable for publication and will be formally accepted for publication once it meets all outstanding technical requirements.

Kind regards,

Fadi Aljamaan

Academic Editor

PLOS ONE

Additional Editor Comments (optional):

Reviewers' comments:

Reviewer's Responses to Questions

**Comments to the Author**

Reviewer #1: All comments have been addressed

2. Is the manuscript technically sound, and do the data support the conclusions?

Reviewer #1: Yes

3. Has the statistical analysis been performed appropriately and rigorously?

Reviewer #1: Yes

4. Have the authors made all data underlying the findings in their manuscript fully available?

Reviewer #1: Yes

5. Is the manuscript presented in an intelligible fashion and written in standard English?

Reviewer #1: Yes

Reviewer #1: The manuscript was revised severely by the autor(s). The study addresses a critical gap in septic shock management by proposing a refined Blood Pressure Response Index (BPRI) that incorporates phenylephrine, enhancing its clinical utility for real-time hemodynamic assessment.

The analysis leverages a large cohort (7,382 patients) from the MIMIC-IV database, employs multivariable logistic regression, sensitivity analyses, and restricted cubic spline models to validate the L-shaped association between BPRI and mortality, ensuring statistical rigor.

The identification of a nonlinear, inverse relationship (AUC = 0.64) between BPRI and short-term mortality, particularly its synergistic predictive value when combined with SOFA scores, offers actionable insights for risk stratification.

Detailed inclusion/exclusion criteria, comprehensive adjustments for confounders, and sensitivity analyses strengthen the validity of results. Ethical approval and data accessibility via MIMIC-IV are appropriately addressed.

The manuscript meets PLOS ONE’s criteria for publication as it presents a scientifically valid, ethically conducted study with clear clinical significance. The methods are transparent, results are robustly analyzed, and limitations (e.g., retrospective design, single-center data) are candidly discussed. The figures and tables are well-structured, though final formatting adjustments (e.g., axis labels, resolution) may be required per journal guidelines. Minor language polishing is recommended to ensure clarity, but the manuscript’s scientific rigor and contribution to sepsis prognostication align with PLOS ONE’s scope. It is suggested that the manuscript be acceptable for submission.

**Do you want your identity to be public for this peer review?** For information about this choice, including consent withdrawal, please see our Privacy Policy

Reviewer #1: **Yes: ** Feng SHEN

---

## [Editor Report · Acceptance letter]

PONE-D-24-45268R2

PLOS ONE

Dear Dr. Xu,

I'm pleased to inform you that your manuscript has been deemed suitable for publication in PLOS ONE. Congratulations! Your manuscript is now being handed over to our production team.

Kind regards,

on behalf of

Dr. Fadi Aljamaan

Academic Editor

PLOS ONE